# Ethanol-Mediated Stress Promotes Autophagic Survival and Aggressiveness of Colon Cancer Cells via Activation of Nrf2/HO-1 Pathway

**DOI:** 10.3390/cancers11040505

**Published:** 2019-04-10

**Authors:** Cesare Cernigliaro, Antonella D’Anneo, Daniela Carlisi, Michela Giuliano, Antonella Marino Gammazza, Rosario Barone, Lucia Longhitano, Francesco Cappello, Sonia Emanuele, Alfio Distefano, Claudia Campanella, Giuseppe Calvaruso, Marianna Lauricella

**Affiliations:** 1Department of Biomedicine, Neurosciences and Advanced Diagnostics (BIND), Institute of Biochemistry, University of Palermo, 90127 Palermo, Italy; cesare.cernigliaro@unipa.it (C.C.); daniela.carlisi@unipa.it (D.C.); sonia.emanuele@unipa.it (S.E.); 2Department of Biological, Chemical and Pharmaceutical Sciences and Technologies (STEBICEF), Laboratory of Biochemistry, University of Palermo, 90127 Palermo, Italy; antonella.danneo@unipa.it (A.D.); michela.giuliano@unipa.it (M.G.); giuseppe.calvaruso@unipa.it (G.C.); 3Department of Biomedicine, Neurosciences and Advanced Diagnostics (BIND), Institute of Human Anatomy, University of Palermo, 90127 Palermo, Italy; antonella.marino@hotmail.it (A.M.G.); rosario.barone@unipa.it (R.B.); francapp@hotmail.com (F.C.); claudiettacam@hotmail.com (C.C.); 4Euro-Mediterranean Institute of Science and Technology, 90100 Palermo, Italy; 5Department of Biomedical and Biotechnological Sciences, University of Catania, I-95123 Catania, Italy; lucia.longhitano@unict.it (L.L.); distalfio@gmail.com (A.D.)

**Keywords:** colon cancer cells, ethanol, Nrf2, HO-1, ER stress, autophagy, MMPs

## Abstract

Epidemiological studies suggest that chronic alcohol consumption is a lifestyle risk factor strongly associated with colorectal cancer development and progression. The aim of the present study was to examine the effect of ethanol (EtOH) on survival and progression of three different colon cancer cell lines (HCT116, HT29, and Caco-2). Our data showed that EtOH induces oxidative and endoplasmic reticulum (ER) stress, as demonstrated by reactive oxygen species (ROS) and ER stress markers Grp78, ATF6, PERK and, CHOP increase. Moreover, EtOH triggers an autophagic response which is accompanied by the upregulation of beclin, LC3-II, ATG7, and p62 proteins. The addition of the antioxidant N-acetylcysteine significantly prevents autophagy, suggesting that autophagy is triggered by oxidative stress as a prosurvival response. EtOH treatment also upregulates the antioxidant enzymes SOD, catalase, and heme oxygenase (HO-1) and promotes the nuclear translocation of both Nrf2 and HO-1. Interestingly, EtOH also upregulates the levels of matrix metalloproteases (MMP2 and MMP9) and VEGF. Nrf2 silencing or preventing HO-1 nuclear translocation by the protease inhibitor E64d abrogates the EtOH-induced increase in the antioxidant enzyme levels as well as the migration markers. Taken together, our results suggest that EtOH mediates both the activation of Nrf2 and HO-1 to sustain colon cancer cell survival, thus leading to the acquisition of a more aggressive phenotype.

## 1. Introduction

Colorectal cancer (CRC) is one of the most widespread cancers in the world [1]. Epidemiological data show the highest incidence of this tumor in countries characterized by high indices of development, such as Australia, Europe, and North America [2]. This geographic variability has been linked to differences in environment and lifestyle [2]. Numerous risk factors have been correlated with the development of CRC, including genetic factors, inflammation, intestinal microflora composition, as well as harmful lifestyle habits, such as smoking, high consumption of red meats, and alcohol intake [3].

Chronic and heavy alcohol consumption, a common lifestyle habit of developed countries, has been associated with an increased risk of developing gastrointestinal cancers, including CRC [4,5]. The association between alcohol drinking and CRC is dose-dependent. A recent meta-analysis of literature from 1996 to 2013 shows that compared with non-occasional drinkers, the relative risk was 0.97 for subjects who drink low doses of ethanol (≤12.5 g/day), 1.04 for moderate drinkers (12.6–49.9 g/day), and 1.21 for heavy drinkers (≥50 g/day) [6]. Alcohol’s influence can also vary based on individual differences in enzymes involved in alcohol metabolism. For example, the presence of ADH1B [7] or ADH1C*1—two polymorphisms of alcohol dehydrogenase (ADH)—has been associated with an increased risk of CRC [8]. In addition to act as a risk factor for carcinogenesis, many studies indicate that chronic alcohol consumption also promotes colon tumor progression [6,7,8,9].

The mechanisms proposed to explain the role of alcohol in CRC promotion include acetaldehyde mutagenic effects, oxidative stress increase, and folic acid deficiency [10,11]. Acetaldehyde, which is produced by ADH, cytochrome P450 2E1 (CYP2E1), or catalase, is considered the most potent carcinogenic metabolite of ethanol [12]. It is highly reactive and mutagenic by forming protein and DNA adducts, which result in DNA mutations and reactive oxygen species (ROS) production [10]. In addition to acetaldehyde, ROS have been also shown to contribute to colon carcinogenesis [13]. In alcoholics, ROS are predominantly generated through CYP2E1, which is induced by chronic alcohol consumption in the mucosa of the esophagus and colon [14]. CYP2E1-induced ROS production has been correlated with the generation of lipid peroxidation products, such as 4-hydroxynonenal and malondialdehyde. These compounds can bind to DNA forming etheno-DNA adducts which are highly carcinogenic [14,15].

It is well known that chronic inflammation also promotes colorectal cancer carcinogenesis. To this regard, it has been reported that the persistent exposure of enterocytes to an inflammatory environment induces molecular alterations favoring tumor development [16]. However, following chronic inflammation epithelial cells can activate adaptive mechanisms to reduce the deleterious action of oxidative stress and survival [17]. The major mechanism by which cancer cells increase their antioxidant proteins is through the activation of nuclear factor erythroid 2-related factor 2 (Nrf2), a transcription factor involved in the transcription of antioxidant and detoxifying factors in response to oxidative stress [18,19]. Nrf2 is normally sequestered in the cytoplasm in an inactive complex with kelch-like ECH-associated protein 1 (Keap1). ROS induce oxidation of critical cysteines of Keap1 and its dissociation from Nrf2 [20]. This event allows Nrf2 to migrate into the nucleus and form transcriptionally active complexes with other proteins, such as Mafs (musculoaponeurotic fibrosarcoma proteins). Such molecular cascades lead to the increase in the transcription of cytoprotective and antioxidant genes, such as superoxide dismutase (SOD), catalase, and heme oxygenase (HO-1), thus favoring cancer cell survival [21]. In addition, previous reports showed that Nrf2, beyond its antioxidant role, also exhibits a protumorigenic activity promoting proliferation and sustaining migration and invasiveness of cancer cells [22]. To this regard, overexpression of Nrf2 has been reported in colon cancer cells and has been related with tumor progression and poor prognosis [23]. Similarly, HO-1, the main target of Nrf2, can also exert a double role in cancer [24,25]. In fact, HO-1 may counteract ROS-mediated carcinogenesis by favoring heme breakdown into biliverdin; however, its overexpression has been shown to provide tumor cells with a more aggressive survival phenotype [26]. Finally, under oxidative stress conditions HO-1 translocates into the nucleus, where it interacts with Nrf2 preventing its GSK3β-mediated phosphorylation coupled with ubiquitin–proteasomal degradation, thereby prolonging Nrf2 nuclear accumulation [27].

The aim of the present research was to evaluate the role of Nrf2/HO-1 axis in promoting colon cancer survival and progression under ethanol stimulation. Our data suggest that ethanol induces both Nrf2 and HO-1 nuclear translocation in response to oxidative and endoplasmic reticulum (ER) stress activation and that this event confers resistance to oxidative stress and contributes to the acquisition of a more invasive phenotype.

## 2. Results

### 2.1. Effects of Ethanol in Colon Cancer Cell Viability

To evaluate the effect of ethanol (EtOH) on colon cancer cells, different colon cancer cell lines (HCT116, HT29, and Caco-2 cells) were exposed to a range of EtOH concentrations (30–300 mM) for different durations. Then, the viability was assessed by 3-(4,5-dimethyl- thiazol-2-yl)-2,5-diphenyltetrazolium bromide (MTT) assay which evaluates the mitochondrial dehydrogenase activity, as reported in methods. As shown in Figure 1A, compared with untreated cells, EtOH, also at high doses, displayed a scarce effect on cell viability at 72 h of treatment, but slightly reduced cell viability at 24 and 48 h, in particular in HT29 and Caco-2 cells. Our data also demonstrated that staining the cells with propidium iodide (PI), a membrane impermeant dye that is generally excluded from viable cells, evidenced that EtOH treatment did not induce cell death in this condition, in which we observed a reduction of cell viability. Thus, the reduced viability assessed by MTT assay in EtOH-treated cells for 24 or 48 h could be due to cell proliferation arrest or reduced metabolic activity of cells.

Morphological observations of cells confirmed the absence of cytotoxicity in all the conditions tested also prolonging EtOH treatment (1 week), leading to the conclusion that colon cancer cells survive even under treatment with high doses of ethanol.

To explore the ability of EtOH to counteract the production of colonies, HCT116 cells were plated with or without the addition of different doses of EtOH, and maintained in culture for additional 14 days to allow the formation of colonies. The results demonstrated no significant difference between control and EtOH-treated cells (Figure 1B), thus suggesting that the compound does not affect colony formation.

To evaluate the biochemical mechanisms linking high doses of alcohol consumption and colon cancer progression, the next experiments were performed using100 mM and 300 mM EtOH, two different high concentrations of alcohol.

### 2.2. Ethanol Treatment Induces Oxidative Stress in Colon Cancer Cells

EtOH metabolism produces acetaldehyde and reactive oxygen species (ROS), which have been correlated with the toxic effects of the compound [28]. To ascertain ROS generation in our model system, we performed fluorescence microscopy analysis by employing the fluorochrome H2DCFDA—a general indicator of intracellular ROS levels. As shown in Figure 2A, EtOH treatment rapidly increases the number of green fluorescent cells, which are indicative of intracellular ROS production. The effect appears at 10 min of exposure to EtOH, then rapidly decreases with the time of treatment. 

Cyclooxygenase 2 (COX2) and inducible nitric oxide synthase (iNOS) are two important enzymes expressed in response to a variety of stimuli that mediate inflammatory processes and tumor progression [29]. Western blotting analysis shows that COX2 and iNOS proteins are significantly up-regulated in EtOH-treated HCT116 cells. In particular, treatment for 3 h with 300 mM EtOH increases the levels of the enzymes respect to untreated control by 2.7 and 2.2 fold, respectively (Figure 2B, *p* < 0.05). 

Moreover, we evaluated the effect of EtOH on the levels of Hsp90 and Hsp60, two heat shock proteins induced by cellular stress [30,31]. As shown in Figure 2B, the exposure to 100 and 300 mM EtOH for 3 h induces a significant increase of both Hsp90 and Hsp60 compared to the untreated condition (*p* < 0.05).

### 2.3. Ethanol Treatment Induces ER Stress in Colon Cancer Cells

Oxidative stress can be responsible for accumulation of unfolded-proteins and induction of ER stress [32]. To explore the ability of EtOH to activate ER stress in colon cancer cells, we analyzed the levels of glucose-regulated protein of 78 kDa (GRP78), which is a chaperone interacting with misfolded proteins and an indicator of ER stress, and CCAAT/enhancer-binding protein homologous protein (CHOP), a transcriptional factor which promotes apoptosis under prolonged ER stress [32]. Data reported in Figure 2C show that, in HCT116 cells, EtOH significantly upregulates the levels of Grp78 after 24 h (*p* < 0.05). In addition, EtOH upregulates CHOP level already after 3 h. After 24 h its level remains higher (2.2-fold, *p* < 0.05) in 100 mM EtOH-treated cells than untreated ones, while it lowers (0.8-fold) in cells exposed to 300 mM EtOH (Figure 2C). EtOH increases the levels of ER stress markers also in HT29 and Caco-2 cells, although the effects were observed after a prolonged time of treatment (24 h) (Appendix A).

We then evaluated the activation of ATF6 and PERK, two unfolding protein response (UPR) sensors which promotes adaptive signal transduction under ER stress [32]. When unfolded proteins increase, the full-length ATF6 is cleaved at Golgi into an active fragment (ATF6f), which migrates into the nucleus upregulating the expression of chaperones, including GRP78 and CHOP [32]. Treatment of HCT116 cells with EtOH decreases the ATF6 full-length protein at 3 h and concomitantly increases that of ATF6f, suggesting the activation of the factor. These effects are not observed at 24 h of EtOH treatment (Figure 2C). Moreover, EtOH treatment also promotes an early PERK increase that significantly (*p* < 0.05) occurs already at 3 h of treatment (Figure 2C).

### 2.4. Ethanol Stimulates A Prosurvival Effect Sustained by An Autophagic Flux In Colon Cancer Cells

Under ER stress condition, autophagy can be activated to degrade unfolded/aggregated proteins in order to maintain cell survival [33,34]. To assess whether this process occurs in our system model, the presence of autophagic vacuoles was evaluated using fluorescence microscopy by staining the cells with monodansylcadaverine (MDC). As Figure 3A shows, dot-likes structures appear in the cytoplasm of EtOH-treated HCT116 cells. MDC-positive fluorescent cells are already observed after 24 h of treatment with both 100 and 300 mM ethanol. The addition of the antioxidant N-acetylcysteine (NAC) to EtOH-treated cells markedly reduces the presence of autophagic vacuoles (Figure 3A), suggesting that autophagy is activated in response to oxidative stress as a prosurvival mechanism.

Then, we analyzed the levels of the autophagic markers by western blotting (Figure 3B). Microtubule-associated protein light chain 3 (LC3) can be present in two different forms: a cytosolic form (LC3-I) and an active lipidated form (LC3-II) [35], which is bound to the autophagosome membrane [36]. As shown in Figure 3B, in HCT116 cells EtOH treatment favors the conversion of LC3-I to LC3-II.

In addition to LC3, the level of beclin—a protein which plays an essential role in autophagosome formation—was evaluated. Data reported in Figure 3B demonstrate that beclin protein level is significantly higher after 3 h of treatment with 300 mMEtOH compared to untreated control (*p* < 0.05). The level remains elevated after 24 h.

Moreover, EtOH treatment increases the level of ATG7, an essential regulator of autophagosome assembly [37]. As shown in the same Figure 3B, ATG7 level significantly (*p* < 0.05) increased after 24 h treatment with 100 mM EtOH with respect to the untreated sample (Figure 3B).

Finally, we also examined the level of p62, a multifunctional protein considered as a marker of autophagic flux. p62 is localized to the autophagosome via LC3 interaction and is constantly degraded by the autophagy–lysosome system [38]. The analysis of p62 shows that EtOH increases the content of this protein at 3 h of treatment, while its level lowers after 24 h (Figure 3B), thereby suggesting that HCT116 cells treated with ethanol undergo to a complete autophagic process.

### 2.5. Ethanol Activates Nrf2-Dependent Antioxidant Pathway

Despite the activation of oxidative and ER stress by EtOH, MTT cell viability and clonogenic assay, reported in Figure 1, show that colon cancer cells survive even after prolonged treatment with high doses of EtOH, thus suggesting the activation of an antioxidant response.

Nrf2 is one of the major transcription factors that promotes cellular defense against oxidative stress. Nrf2 is maintained in an inactive cytosolic complex with Keap1 [39]. Our data show that Nrf2 level is enhanced in HCT116 cells treated with EtOH. In fact, in comparison with control cells the level of Nrf2 protein increases with 300 mM EtOH approximately by 1.5-fold (*p* < 0.05) after 3 h of treatment (Figure 4). Data reported in the same Figure 4, also demonstrate that the increase of Nrf2 is accompanied by a significant decrease of Keap1 in cells treated for 3 h with 300 mM EtOH compared to untreated control (*p* < 0.05).

It is known that Nrf2 controls the transcription of many antioxidant and detoxifying genes, such as superoxide dismutase (MnSOD), catalase, and heme oxygenase (HO-1) [21]. We demonstrated that the levels of these proteins transcriptionally regulated by Nrf-2 markedly increase after 24 h of treatment with 300 mMEtOH (Figure 4). In particular, the increase in the presence of 300 mM EtOH is estimated 1.4-fold for catalase, 4-fold for MnSOD (*p* < 0.001), and 5.5-fold for HO-1 (*p* < 0.001) after 24 h of treatment (Figure 4). An increase in the levels of Nrf-2 and HO-1 is also observed in both HT29 and Caco-2 cells after EtOH treatment (Appendix A).

### 2.6. Ethanol Promotes Nrf2 and HO-1 Translocation

Then we analyzed if the Nrf2 increase is accompanied by its nuclear translocation. Western blotting analysis showed that Nrf2 is present in both nuclear and cytosolic fraction in untreated HCT116 cells and that EtOH treatment increases the nuclear level of Nrf2, while, concomitantly, decreases its cytosolic fraction (Figure 5A). The analyses by confocal microscopy confirmed that in control cells Nrf2 is preferentially distributed in the cytoplasm, while in treated cells the immunoreactivity is mainly nuclear (Figure 5B).

It has been reported that HO-1 can undergo to intramembrane proteolysis and translocation into the nucleus to sustain tumor survival and invasiveness without depending on its enzymatic activity [40]. Interestingly, western blotting analyses showed that EtOH is able to increase nuclear HO-1 levels (Figure 5A). This effect was already observed after 3h of treatment with 300 mM EtOH (*p* < 0.01). Nuclear HO-1 translocation in EtOH-treated cells was also confirmed by confocal microscopy experiments (Figure 5B, white arrows).

### 2.7. Ethanol Increases MMPs and VEGF in Colon Cancer Cells

Given that metalloproteases (MMPs) are critical to cell invasion and metastasis [41], the expression and activity of MMP2 and MMP9 in EtOH-treated HCT116 cells were examined by both Western blotting analysis and gelatin zymography. Our results show that there is a significant increase in the protein level of active MMP2 (7-fold, *p* < 0.001) in HCT116 cells treated with 300 mM EtOH already after 24 h of treatment (Figure 6A). A minor increase (2.2-fold, *p* < 0.01) is observed for active MMP9 (Figure 6A). Moreover, gelatin zymography showed that 300 mM EtOH also increases the activity of both metalloproteinases (Figure 6B).

Finally, we also demonstrated that EtOH increases the level of vascular endothelial growth factor (VEGF) in HCT116 cells. As shown in Figure 6A, in comparison with untreated cells, the level of VEGF protein significantly increases after 24 h in EtOH-treated HCT116 cells approximately by 3.5-fold (*p* < 0.01) with 100 mM and by 3.2 fold (*p* < 0.01) with 300 mM, respectively.

### 2.8. Role of Nrf2/HO-1 Axis in Colon Cancer Survival and Aggressiveness

To further demonstrate the contribution of Nrf2 in protecting colon cancer cells by oxidative damage induced by EtOH, Nrf2 was downregulated by transfecting HCT116 cells with a specific siRNA pool directed against Nrf2. Western blotting analysis showed that the levels of Nrf2 protein in siNrf2-transfected group are significantly decreased (*p* < 0.05) after 24 h compared with the siRNA control group (Figure 7). Interestingly, the increases in HO-1 and MnSOD protein level caused by 300 mM EtOH treatment is suppressed by Nrf2 siRNA (Figure 7A), thus confirming that the expression of such antioxidant proteins is under Nrf2 control. Moreover, cell viability assay showed that Nrf2 silencing reduces HCT116 cell viability (Figure 7B), thus suggesting a protective role of Nrf2 axis against EtOH-mediated toxic effect.

Furthermore, to evaluate whether also nuclear HO-1 translocation favors the survival of EtOH-treated colon cancer cells, HCT116 cells were incubated in the presence of E64d, an inhibitor of the protease responsible for the proteolytic cleavage of HO-1, an event necessary for its nuclear translocation [42]. Combination of E64d with 300 mM EtOHis accompanied by a significant reduction in nuclear localization of HO-1 (*p* <0.05) (Figure 8A). In addition, the data showed that inhibition of nuclear translocation of HO-1 by E64d significantly reduces the EtOH-induced increase of MnSOD (Figure 8B), thus supporting the conclusion that nuclear HO-1 could regulate MnSOD expression.

It has been reported that both Nrf2 and HO-1 activation promote tumor progression by regulating proinvasive and angiogenetic factors [22,43]. Thus, further analyses were performed to examine whether the Nrf2/HO-1 axis also plays a role in the acquisition of a tumor aggressive phenotype in EtOH-treated colon cancer cells. As reported in Figure 7 and Figure 8, in HCT116 cells the increase in MMP2 protein level induced by EtOH treatment at 300 mM is significantly (*p* < 0.05) suppressed by both Nrf2 siRNA (Figure 7A) and E64d (Figure 8B). Moreover, the increase in VEGF level in EtOH-treated cells is reduced by Nrf2 silencing (Figure 7A) and prevented by E64d (Figure 8B). These data indicate that activation of Nrf2/HO-1 axis could enhance colon cancer progression by inducing MMP2 and VEGF expression.

## 3. Discussion

Epidemiological studies highlighted that heavy alcohol drinking promotes colon cancer progression [44], although the underlying molecular mechanisms are still not clear. Data reported in this paper show that colon cancer cells survive even under treatment with high doses of ethanol (100–300 mM), which have been shown to exert toxic effects in other tumor cell lines [45,46]. It is interesting to note that concentrations ranging from 10 to 100 mM correspond to blood levels in humans that could result from moderate-to-heavy alcohol drinking [47].

Supportive signs of evidence suggest that EtOH is able to increase ROS level in different cell lines, including colon cancer cells, through both ADH and CYP2E1 activity [48]. Moreover, chronic EtOH exposure causes inflammation in different organs, like pancreas and liver, as indicated by the increase in proinflammatory cytokines and chemokines [49,50]. In line with these observations, we demonstrated that EtOH stimulates oxidative stress and an inflammatory response in colon cancer cells. This conclusion is supported by several pieces of evidence, such as rapid ROS production, the increase in the levels of two enzymatic markers of inflammation iNOS and COX2, and the upregulation of Hsp90 and Hsp60 induced by EtOH in colon cancer cells.

Our results also provided evidence that high doses of EtOH trigger ER stress as demonstrated by upregulation of the ER stress markers Grp78, CHOP, and PERK, as well as the increase in the active form of the transcription factor ATF6. These results are in line with the observation that ER stress contributes to alcoholic damage of major organs such as liver, pancreas, and brain [51,52].

Accumulating evidence suggest that activation of autophagy, which degrades proteins in organelles such as mitochondria and the ER, can play a protective role against the toxic effects of ER stress [53]. On the other hand, ER stress can activate autophagy and there is a considerable cross-talk between the ER and autophagy [54]. It has been reported that ER stress-activated PERK promotes the phosphorylation of eIF2α to induce the activation of LC3, an autophagosome marker [55]. Moreover, Shimodaira et al. [33] demonstrated that in colon cancer cells activation of ER marker CHOP promotes autophagy by activating inositol-requiring enzyme 1α (IRE1α). Our results provide evidence that EtOH induced autophagy in colon cancer cells as demonstrated by the augmented accumulation of acidic intracellular compartments. This effect is prevented by the addition of the antioxidant N-acetylcysteine, thus suggesting that autophagy is activated as a survival mechanism in response to oxidative stress. Our results are in line with the observation that EtOH activates autophagy in neuronal cells to prevent the oxidative damage of ROS [56]. The activation of autophagy in our system was also confirmed by the observation that EtOH promoted the cleavage of the cytosolic form of LC3-I to LC3-II, as well increasing the level of beclin and ATG7, two factors involved in the autophagosome formation [37]. Moreover, the observation that the level of p62 protein, a marker of autophagic degradation, decreases after 24 h of EtOH treatment, which suggested to us that the compound is able to trigger a complete autophagic flux.

In response to oxidative and ER stress production, we showed a significant activation of Nrf2, a transcription factor which acts as a key regulator of antioxidant-responsive genes [57]. Under our experimental conditions, EtOH promotes nuclear Nrf2 translocation, as suggested by the increase of nuclear Nrf2 content already at 3 h following EtOH treatment. This event is a consequence of the decrease of Keap1, a protein which in unstressed conditions sequesters Nrf2 in a cytoplasmic complex leading to its ubiquitination and consequent proteasomal degradation [58]. Nuclear translocation of Nrf2 by EtOH could be promoted by oxidative events, in line with the observation that induction of oxidative stress triggers Nrf2 nuclear import through the oxidation of redox-sensitive cysteines within Keap1 [59]. Moreover, nuclear translocation could be also favored by ER stress activation. In fact, it has been shown that phosphorylation of Nrf2 by PERK—a kinase activated following the accumulation of unfolded proteins in ER—promotes dissolution of Nrf2/Keap1 complexes and Nrf2 nuclear import [60].

Activation of Nrf2 is responsible for transcription of a battery of genes encoding antioxidant enzymes such as MnSOD, catalase, and HO-1 [61]. Our data indicated that the level of both MnSOD and HO-1 is significantly upregulated after EtOH treatment in colon cancer cells, suggesting a protective role of these enzymes against EtOH-induced oxidative stress. The observation that Nrf2 silencing significantly reduces EtOH-induced HO-1 and MnSOD increase suggest to us that this effect is mediated by the activation of Nrf2 transcriptional activity.

Interestingly, our data also provide evidence that EtOH promotes HO-1 nuclear translocation already after 3 h of treatment. Nuclear expression of HO-1 has been detected in several tumors and it has been correlated with tumor growth and invasion [43]. Our data showed that inhibition of HO-1 nuclear translocation by the protease inhibitor E64d significantly reduces EtOH-induced MnSOD increase, suggesting that HO-1 nuclear translocation could cooperate with Nrf2 to stimulate antioxidant response and colon cancer cell survival. This suggestion is in accordance with some recent evidence that demonstrates that nuclear HO-1 modulates the activation of Nrf2, leading to activation of antioxidant genes [27].

Activation of the Nrf2/HO-1 axis represents a double-edged sword in cancer [61]. Increase in antioxidant enzymes by Nrf2 prevents the development of tumors by counteracting the genotoxic damage induced by ROS [62]. In line with this consideration, several dietary phytochemicals exert a cancer preventive effect by activating the Nrf2/HO-1 axis [63,64]. Moreover, activation of Nrf2 reduces chronic inflammation which has been correlated with CRC induction [65]. On the other hand, activation of the Nrf2/HO-1 antioxidant response in tumor cells can promote tumor survival by creating an optimal environment for cell growth [61]. Overexpression of Nrf2 has been detected in primary CRC and metastatic tissues relative to normal colon and contributes to chemoresistance in CRC cell lines [66]. In addition, it has been reported that Nrf2 increases CRC risk by promoting angiogenesis and uncontrolled proliferation [67]. Moreover, HO-1 overexpression has been associated with a more aggressive behavior of tumors and poor prognosis in various cancers [68,69]. Western blotting and zymography analyses demonstrated that EtOH increases both the levels and the activity of MMP-2 and 9—two enzymes involved in tumor progression—and upregulates VEGF, the main angiogenetic factor. Interestingly, Nrf2 silencing prevented EtOH-induced MMP-2 increase and reduced that of VEGF. Moreover, inhibition of HO-1 nuclear translocation by E64d counteracted the effect of EtOH on both MMP-2 and VEGF, thus markedly suggesting a role of Nrf2/HO-1 axis in colon cancer progression.

Collectively, our findings demonstrate that high doses of EtOH enhance autophagy and activation Nrf2/HO-1 axis in colon cancer cells. These events sustain the survival of cancer cells protecting them from oxidative and ER stress induced by EtOH. Moreover, we show, for the first time, that the activation of Nrf2/HO-1 axis could be also responsible for colon cancer progression through the acquisition of a metastatic behavior, as demonstrated by the increase in the levels of MMP-2 and VEGF. A schematic model of EtOH effects on colon cancer cells is shown in Figure 9. This study provides a novel mechanistic link between ethanol-induced activation of Nrf2/HO-1 pathway and increased survival and aggressiveness of colon cancer cells in in vitro models. However, to open to this new scenario, further in vivo investigations are strongly required to sustain the role of Nrf2/HO-1 activation in CRC progression. Detection of Nrf2 and HO-1 overexpression in CRC biopsies of alcohol drinkers could be, indeed, used as potential novel biomarkers to monitor CRC progression.

## 4. Materials and Methods

### 4.1. Cell Cultures and Chemicals

The human colon cancer lines HCT116, HT29, and Caco-2 (Interlab Cell Line Collection, ICLC, Genova, Italy) were grown in monolayer in flasks of 75 cm^2^ in RPMI 1640 medium, supplemented with 10% (*v*/*v*) heat-inactivated fetal bovine serum (FBS), 2 mM L-glutamine, 100 U/mL penicillin, and 50 µg/mL streptomycin in a humidified atmosphere of 5% CO_2_ in air at 37 °C. To study the effects of ethanol (EtOH), cells were detached using trypsin-EDTA (0.5 mg/mL trypsin and 0.2 mg/mL EDTA) and plated in accordance to the experimental conditions, as described in the paragraphs below. Cells were allowed to adhere for 24 h and then treated with different concentrations of EtOH at different times.

All the reagents used for cell cultures were purchased from Euroclone (Pero, Italy). EtOH, E64d and all chemicals, except when stated otherwise, were supplied by Sigma-Aldrich (Milan, Italy).

### 4.2. Cell Viability

To evaluate the effect of EtOH on cell viability the MTT (3-(4,5-dimethylthiazol-2-yl)-2,5-diphenyl tetrazolium bromide) colorimetric assay was used as previously reported [70]. In brief, HCT116, HT29, and Caco-2 cells (7 × 10^3^/200 µL/well) were plated in 96-wells and treated with various concentrations of EtOH (30–300 mM) for different times. Fresh ethanol-containing medium was added to cells daily. Then, 20 μL MTT (11 mg/mL) was added and cells were incubated at 37 °C for 4 h. Finally, the medium was removed and 100 µL of lysis buffer (20% sodium dodecyl sulfate in 50% N,N-dimethylformamide) was added. At the end, the absorbance of the formazan was measured directly at 490 nm with 630 nm as a reference wavelength using an automatic ELISA plate reader (OPSYS MR, Dynex Technologies, Chantilly, VA, USA). Cell viability was expressed as the percentage of the OD value of EtOH-treated cells compared with untreated samples used as control. Each experiment was performed in triplicate. The viability of cells was also assessed through the use of propidium iodide (PI) dye exclusion assay. The intact membrane of live cells excludes a variety of dyes that easily penetrate the damaged, permeable membranes of dead cells. After incubation with ethanol, 2 μg/mL PI was added and the incubation was protracted for 15 min before the visualization of cell morphology by a Leica DC 300F microscope (Leica microsystems, Wetzlar, Germany) equipped with a rhodamine filter (excitation wavelength of 596 nm and emission wavelength of 620 nm). Also, cell morphology was visualized using a Leica DC 300F microscope inverted.

### 4.3. Clonogenic Assay

For the colony formation assay, a single cell suspension (200 cells/well) was plated into each well of a 6-well plate and incubated at 37 °C for 2 weeks. The colonies were fixed and stained with a dye solution containing crystal violet as reported by Raffa et al. [71]. All assays were replicated three times.

### 4.4. Western Blotting Analysis

Cell lysates were prepared as reported [72] and protein concentration was determined by Bradford Protein Assay (Bio-Rad Laboratories S.r.l., Segrate, Milan, Italy). Protein extracts (30 µg/sample) were subjected to SDS-polyacrylamide gel electrophoresis. Then, proteins were blotted on nitrocellulose membranes. Primary antibodies used for the identification of catalase, Lamin B, MnSOD, Nrf2, iNOS, COX2, Grp78, Chop, MMP2, MMP9, and PERK (1:200) were purchased from Santa Cruz Biotechnology (St. Cruz, CA, USA); β-actin (1:1000) from Sigma Aldrich; Hsp60, Hsp90, and HO-1 (1:1000) from Enzo Life Sciences, (Milan, Italy), ATG7 (1:1000) from Cell Signaling Tecnology (Beverly, MA, USA); and ATF6 (1:300) from Novus Biologicals (Milano, Italy). Membranes were then incubated with HRP-conjugated secondary antibody (1:5000) (Pierce, Thermo Fisher Scientific, Loughborough, UK) and the signals were detected using enhanced chemiluminescence (ECL) reagents (Cyanagen, Bologna, Italy). The signal obtained was visualized and photographed with ChemiDoc XRS (Bio-Rad, Hercules, CA, USA). After analysis, membranes were stripped (200 mM glycine, 0,1% SDS, 1% Tween 20, pH 2.2) and reincubated. The intensity of the bands was performed using Quantity One software (Bio-Rad). After checking that β-actin content was not modified by ethanol treatment vs untreated sample, in accordance with Lamichhane et al. [73], β-actin was used as a loading control and for band normalization. All the blots shown are representative of at least three separate experiments.

### 4.5. Extraction of Cytosolic and Nuclear Fraction

HCT116 cells were seeded in 100-mm tissue culture dishes (1 × 10^6^ cells/5 mL culture medium) and, after treatment with EtOH, were lysed as reported [74]. In particular, cells were washed in PBS and scraped with lysis buffer (250 mM Sucrose, 20 mM HEPES, 10 mM KCl, 1.5 mM MgCl_2_, 1 mM EDTA, 1 mM EGTA, 1 mM DTT, and protease inhibitors, pH 7.4). Next, cells were passed 10 times through a needle of 25 g on ice for 20 min. The homogenates were centrifuged at 1000× *g* for 10 min at 4 °C. The pellets were resuspended in lysis buffer and passed 10 times through a needle of 25 g and centrifuged at 1000× *g* for 10 min at 4 °C. The pellets of the second centrifugation (nuclear fraction) were lysed with RIPA buffer (1% NP-40, 0.5% sodium deoxycholate, 0.1% SDS, inhibitors of proteases: 25 μg/mL aprotinin, 1 mM PMSF, 25 μg/mL leupeptin, and 0.2 mM sodium pyrophosphate) and sonicated. The supernatants obtained from the first centrifugation were centrifuged at 10,000× *g* for 30 min at 4 °C. The supernatants obtained were considered as cytosolic fraction. Nuclear and cytosolic fractions were used to evaluate Nrf2 and HO-1. β-actin and LaminB were used as cytoplasmic and nuclear markers, respectively.

### 4.6. Monodansylcadaverine Test

To evaluate the formation of autophagic vacuoles monodansylcadaverine (MDC) test was employed as reported [75]. HCT116 cells (7 × 10^3^/200 µL culture medium) were plated in 96-wells plates and treated with EtOH. After treatment, cells were incubated with 50 µM MDC for 10 min at 37 °C in the darkness. Then, cells were washed with PBS and analysed by fluorescence microscopy using a Leica DMR (Leica Microsystems, Milan, Italy) microscope equipped with a DAPI filter system (excitation wavelength of 372 nm and emission wavelength of 456 nm). Images were acquired by computer imaging system (Leica DC300F camera, Milan, Italy). Three different visual fields were examined for each condition.

### 4.7. Nrf2 siRNA Transfection

RNA interference of Nrf2 was performed using FlexiTube siRNA (SI03246950, SI03246614 Qiagen, Hilden, Germany) targeting Nrf2. A nonsilencing siRNA (SI03650318, Qiagen) was used as a negative control. As previously reported [76] for transfection, HCT116 cells were seeded (2 × 10^5^ cells/well) in 6-well plates and cultured in antibiotic-free RPMI 1640 medium supplemented with 10% FBS for 24 h to reach approximately 60–80% confluence before transfection. Specific siRNAs (50 nM final concentration) and negative siRNA control (50 nM) were transfected for 5 h into the cells in the presence of 5 μL Lipofectamine 2000 (Invitrogen, Carlsbad, CA, USA) in a final volume of 1 mL serum/antibiotic-free RPMI 1640 medium. The reaction was stopped by replacing the culture medium with complete RPMI 1640 medium. After 24 h of transfection, cells were treated with EtOH for other 24 h or 48 h. Then, cells were employed for immunoblotting analysis or MTT.

### 4.8. Measurement of Intracellular ROS Content

Intracellular ROS production was detected using the cell-permeant 2′,7′-dichlorodihydrofluorescein diacetate (H2DCFDCA Molecular Probes; Eugene, OR, USA) as described [77]. HCT116 cells (7 × 10^3^/200 µL) were seeded in 96-well plates and incubated with 300 mM EtOH for different times. After treatment, cells were washed with PBS and incubated with a 10 μM H2DCFDA for 15 min at 37 °C in the dark. Finally, cells were resuspended in PBS and analyzed by fluorescence microscopy using a Leica DMR (Leica Microsystems S.r.l., Wetzlar, Germany) inverted microscope equipped with a FITC filter system (excitation wavelength of 485 nm and emission wavelength of 530 nm). Images were acquired by computer imaging system (Leica DC300F camera). Three different visual fields were examined for each condition.

### 4.9. Gelatin Zymography

HCT116 cells were seeded in 100-mm tissue culture dishes (5 × 10^5^ cells/ 5 mL culture medium). After 48 h of EtOH treatment, cells were washed in PBS and scraped with lysis buffer. The lysates were centrifuged at 800× *g* for 10 min. The samples (50 μg of proteins prepared in sample buffer: 50 mM Tris-HCl, 2% SDS, 0.1% Bromophenol Blue, 40% Glycerol, pH 6.8) were loaded on polyacrylamide gels (10%) with 10× gelatin and subjected to electrophoresis. After, the gel was washed for 1 h with enzyme renaturing buffer (200 mM NaCl, 5 mM CaCl_2_, 5 μM ZnCl_2_, 2,5% (*v*/*v*) Triton X-100 and 50 mM Tris-HCl, pH 7.5) and incubated overnight at 37 °C with developing buffer (50 mM Tris base, 200 mM NaCl, 5 mM CaCl_2_, pH 7.5). Then, the gel was incubated for 30 min at room temperature with staining solution (0.125% Coomassie brilliant blue R-250, 50% methanol, 20% acetic acid) and washed with destaining solution (30% methanol, 0,01% formic acid) until clear bands of MMP activity are visible in the blue background.

### 4.10. Immunofluorescence

HCT116 cells (8 × 10^3^) were plated on coverslips and treated with 300 mM EtOH for different times. Immunofluorescence was performed as previously described [78,79]. Briefly, after washing twice in PBS, cells were fixed in methanol for 30 min at room temperature. After fixation, cells were washed three times in PBS for 5 min and treated with a blocking solution (3% BSA in PBS) for 30 min. Subsequently, the cells were washed twice in PBS and incubated with the primary antibody directed against HO-1 (anti-rabbit, Enzo Life Sciences) or against Nrf2 (anti-rabbit, Santa Cruz Biotecnology, St Cruz, CA, USA) at a dilution 1:100, overnight at 4 °C. Then, cells were washed three times in PBS for 5 min and incubated for 1h with a conjugated secondary antibody: anti-rabbit IgG–FITC produced in goat (Sigma-Aldrich) at dilution 1:200. Nuclei were stained with Hoechst Stain Solution (1:1000, Hoechst 33258, Sigma-Aldrich). The images were captured using a Leica Confocal Microscope TCS SP8 (Leica Microsystems). Ten random visual fields were examined for each condition.

### 4.11. Statistical Analysis

Data were represented as mean ± S.E. and analysis was performed using the Student’s *t*-test and one-way analysis of variance. Comparisons between the control (untreated) vs. all treated samples were made. If a significant difference was detected by ANOVA analyses, this was re-evaluated by post-hoc Bonferroni’s test. GraphPadPrismTM 4.0 software (Graph PadPrismTM Software Inc., San Diego, CA, USA) was used for statistical calculations. The statistical significance threshold was fixed at *p* < 0.05.

## 5. Conclusions

In summary, cell culture experiments performed in three different colon cancer cell lines demonstrate that nuclear translocation and consequent activation of both Nrf2 and HO-1 in response to high doses of ethanol exert a protective effect against the toxic effects of alcohol-induced oxidative and ER stress. Moreover, Nrf2/HO-1 activation also favors the acquisition of a more aggressive phenotype through the upregulation of proinvasive and angiogenetic factors like MMPs and VEGF. This might represent a causative link between alcohol consumption and an increased risk of CRC progression.

## Figures and Tables

**Figure 1 cancers-11-00505-f001:**
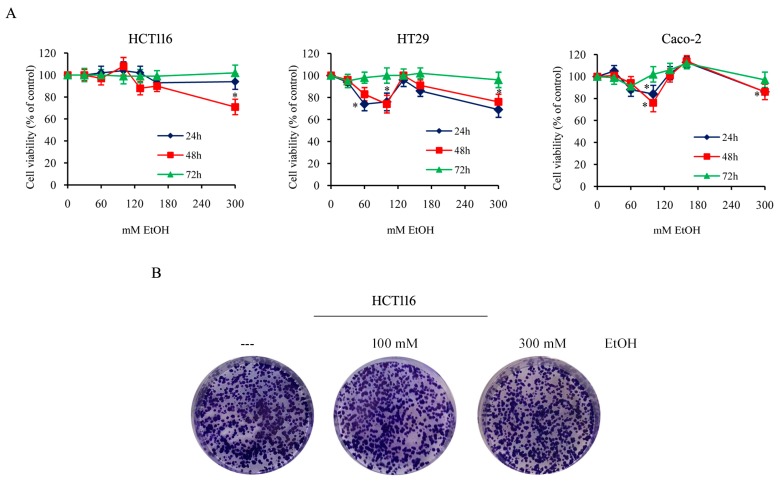
Ethanol effect on colon cancer cell viability and colony generation ability. (**A**) HCT116, HT29, and Caco-2 cells (7 × 10^3^) were incubated in the presence of various doses of EtOH (30–300 mM) for the indicated times. Cell viability was assessed by 3-(4,5-dimethyl- thiazol-2-yl)-2,5-diphenyltetrazolium bromide (MTT) assay and expressed as the percentage of the viable control cells in untreated cultures. Values are the means of three independent experiments ± S.E. (*) *p* < 0.05 compared to the control. (**B**) The clonogenic assay was performed seeding a single cell suspension (200 cells/well) in 6-well plate and after 48 h treating it with different doses of EtOH. Clonogenic ability was evaluated after 14 days. Photographic images of cells after staining with crystal violet were reported. Results are representative of three independent experiments.

**Figure 2 cancers-11-00505-f002:**
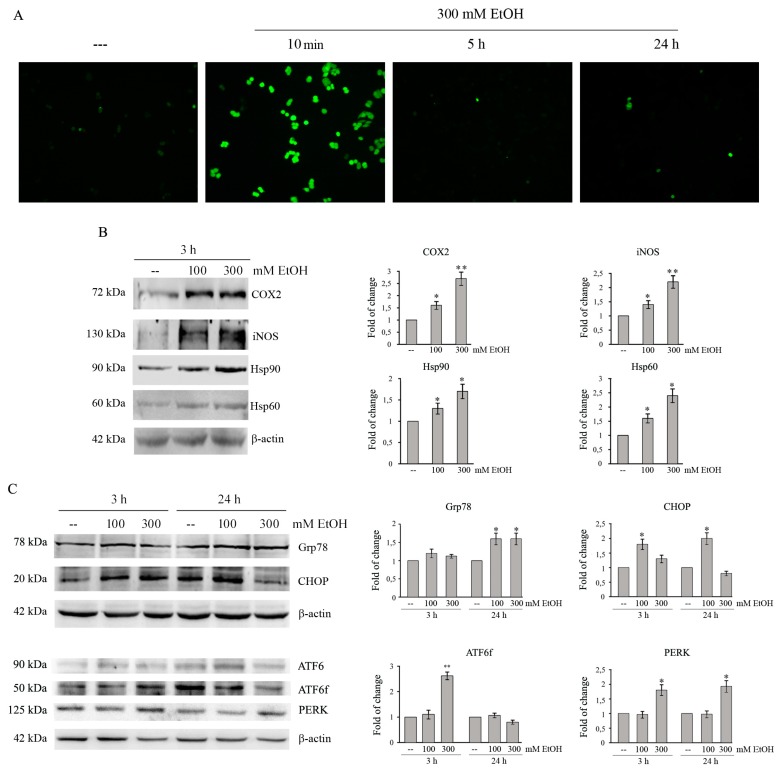
Ethanol induces oxidative and ER stress in colon cancer cells. (**A**) HCT116 cells (7 × 10^3^) were treated with 300 mM EtOH for the indicated times. Reactive oxygen species (ROS) production was assayed by H2DCFDA staining. Images were taken by a Leica DC300F microscope (200× original magnification) using a FITC filter. Results are representative of three independent experiments. (**B**,**C**) Cells (1 × 10^5^) were treated with 100 or 300 mMEtOH for the indicated times. Cell lysates (30 μg) were analyzed by Western blotting for COX2, iNOS, Hsp90, and Hsp60 (**B**) or Grp78, CHOP, ATF6 and its fragmented form (ATF6f) and PERK (**C**). Densitometric analysis of bands was carried out as reported in Material and Methods and data were normalized to β-actin expression. Results are from three independent experiments and statistical significance was determined using one-way ANOVA followed by Bonferroni’s test. (*) *p* < 0.05 and (**) *p* < 0.01 compared to the untreated sample.

**Figure 3 cancers-11-00505-f003:**
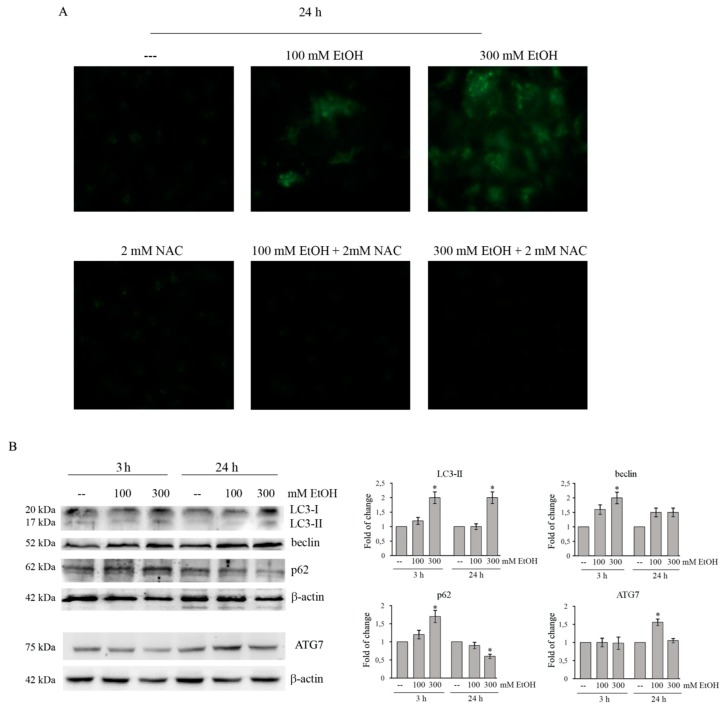
Ethanol induces autophagy in HCT116 cells. (**A**) HCT116 cells (7 × 10^3^) were treated with 100 or 300 mMEtOH for 24 h in the presence or absence of 2 mM NAC. Autophagic vacuoles production was assayed by MDC staining under a Leica DC300F microscope (400× original magnification) using a DAPI filter. Results are representative of three independent experiments. (**B**) Western blotting analysis of LC3, beclin, ATG7, and p62 in HCT116 cells treated for the indicated times with 100 or 300 mM EtOH. Quantitative estimations of the protein levels were determined by densitometry measurements of Western blotting from three independent experiments after normalization with β-actin. (*) *p* < 0.05 compared to the untreated sample.

**Figure 4 cancers-11-00505-f004:**
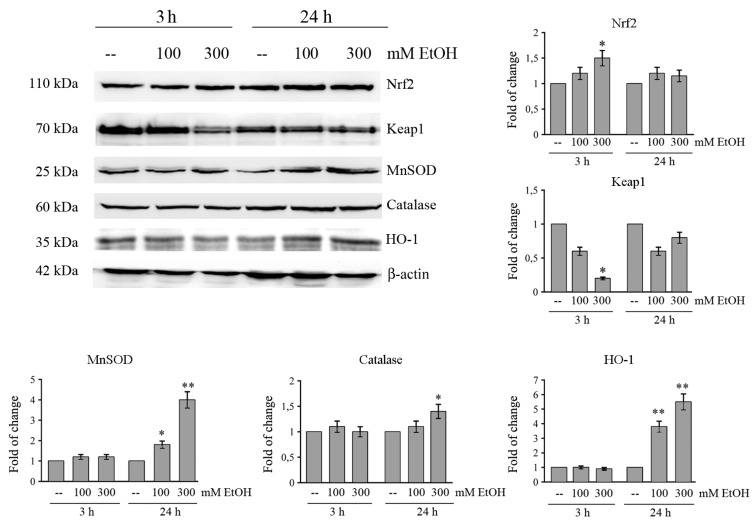
Effects of ethanol on Nrf2-dependent antioxidant pathway in HCT116 cells. Cells (1 × 10^5^) were incubated in the presence of various doses of EtOH for different periods. The levels of Nrf2, Keap1, MnSOD, catalase, and HO-1 were assessed by Western blotting analysis. Immunoblot was quantified by densitometry and normalized against β-actin expression. Results are from three independent experiments and statistical significance was determined using one-way ANOVA followed by Bonferroni’s test. (*) *p* < 0.05 and (**) *p* < 0.01 compared to the untreated sample.

**Figure 5 cancers-11-00505-f005:**
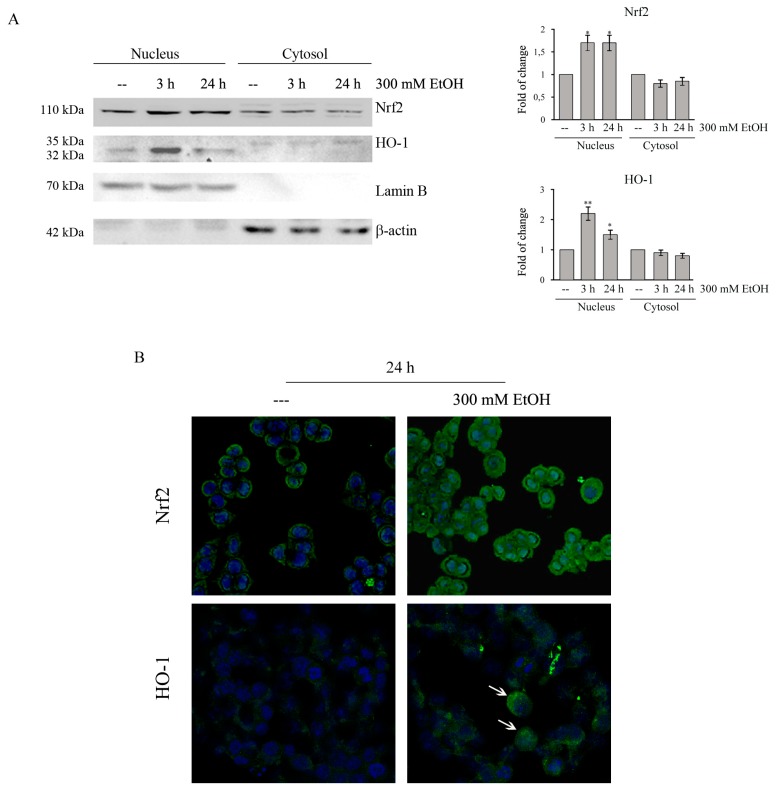
Ethanol induces the nuclear translocation of cytosolic Nrf2 and HO-1 in HCT116 cells. (**A**) Cells were treated for 3 h and 24 h with 300 mM EtOH. Equal amounts of nuclear or cytosolic proteins (30 μg) were analyzed by Western blotting and quantified by densitometry for Nrf2 and HO-1 expression normalized against Lamin B and β-actin. Representative blots of three independent experiments are shown. (*) *p* < 0.05 and (**) *p* < 0.01 compared to the untreated sample. (**B**) HCT116 cells (8 × 10^3^) were grown on coverslips and treated for 24 h with 300 mM EtOH. EtOH-induced nuclear translocation was observed under confocal microscope TCS SP8 employing Nrf2 or HO-1 specific antibodies followed by incubation with a fluorescein isothiocyanate (FITC) conjugated IgG secondary antibody (green). The cells were also stained with Hoechst (blue fluorescence) to visualize nuclear morphology. Original magnification: 400×. The results are representative of three independent experiments.

**Figure 6 cancers-11-00505-f006:**
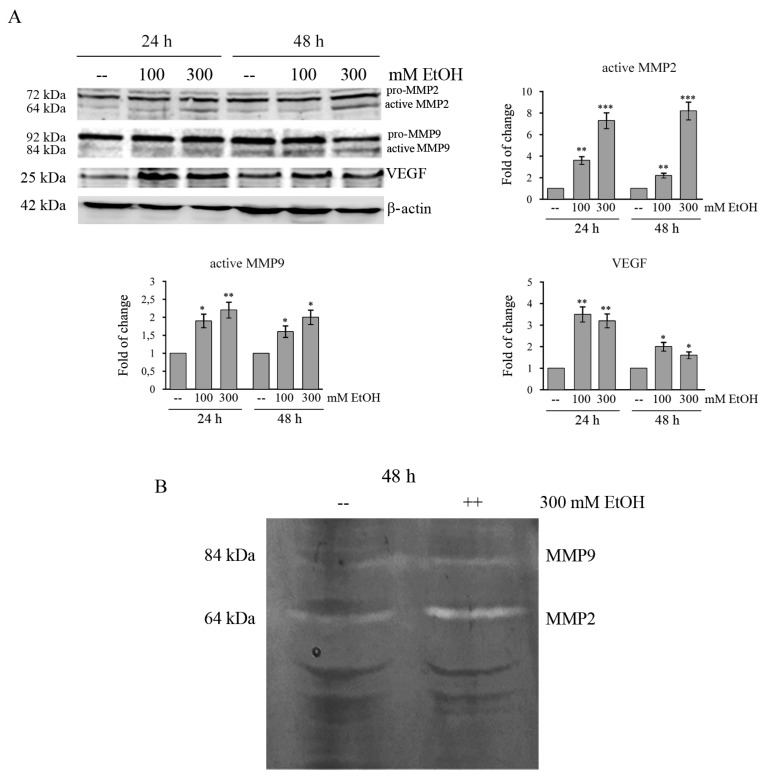
Ethanol increases the levels of VEGF and activates MMPs in HCT116 cells. (**A**) Western blotting analysis of MMP2, MMP9, and VEGF in HCT116 cells treated for different times with 100 or 300 mMEtOH. Quantitative estimations of the protein levels were determined by densitometry measurements of western blotting from three independent experiments after normalization with β-actin. (*) *p* < 0.05, (**) *p* < 0.01, (***) *p* < 0.001 compared to the untreated sample. (**B**) The activity of MMP2 and MMP9 in HCT116 cells treated with 300 mM EtOH for 48 h was examined by gelatin zymography assay. The results are representative of three independent experiments.

**Figure 7 cancers-11-00505-f007:**
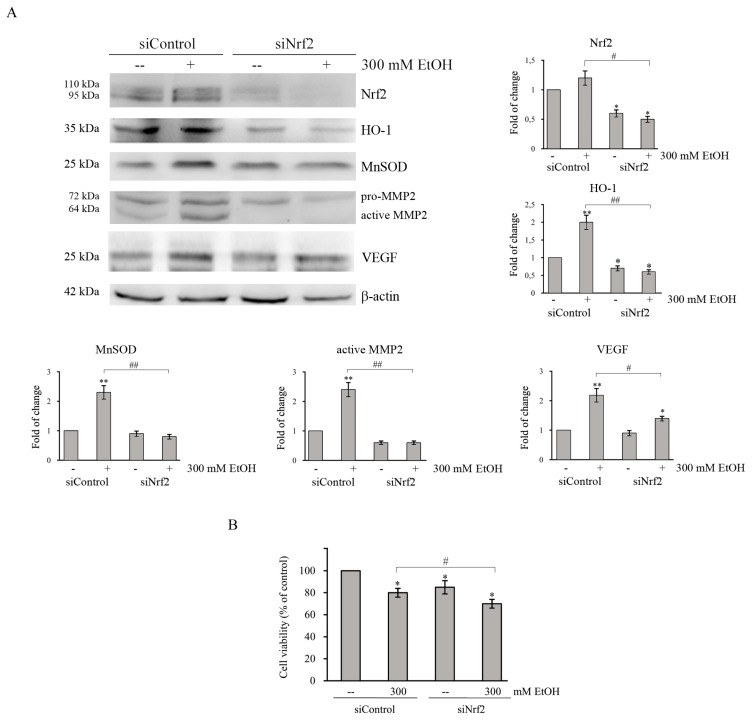
Nrf2-silencing represses the increase in the levels of antioxidant enzymes, MMP2 and VEGF induced by EtOH treatment. (**A**,**B**) HCT116 cells (1 × 10^5^) were transiently transfected with a nonspecific siRNA (siControl) or with Nrf2 specific siRNA pool (siNrf2) and, 24 h after transfection, cells were treated with 300 mM EtOH for additional 24 h or 48 h. (**A**) The levels of HO-1, MnSOD, MMP2, and VEGF were analyzed by Western blotting after 24 h of transfection. The successful of Nrf2-silencing was verified by measuring the level of Nrf2 in transfected cells. Quantitative estimations of the protein levels were determined by densitometry measurements of western blotting from three independent experiments after normalization with β-actin (*) *p* < 0.05 and (**) *p* < 0.01 compared to the untreated sample. (^#^) *p* < 0.05, (^##^) *p* < 0.01. (**B**) Effects of Nrf2-silencing on the viability of HCT116 cells evaluated after 48 h of EtOH treatment by MTT assay and expressed as the percentage of the viable siControl cells in untreated cultures. Values are the means of three independent experiments ± S.E. (^#^) *p* < 0.05 between the two groups.

**Figure 8 cancers-11-00505-f008:**
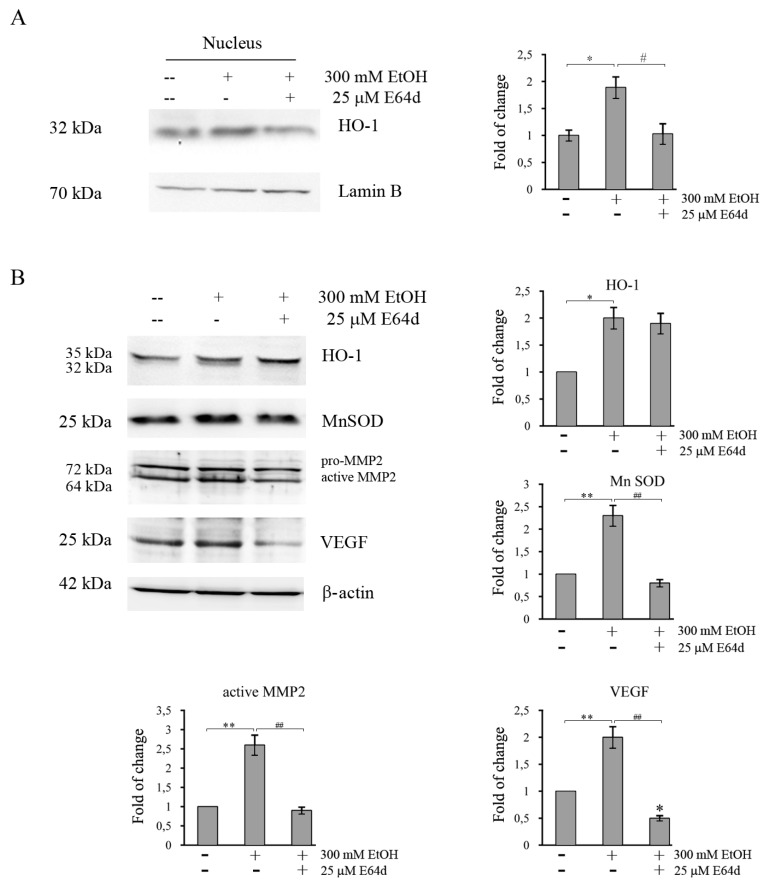
E64d counteracts the increase in antioxidant enzymes, MMP2 and VEGF induced by EtOH treatment in HCT116 cells. Cells were pretreated for 3 h with E64d, then 300 mM EtOH was added and the treatment was protracted for another 24 h. Nuclear and cytosolic fractions were prepared as reported in Materials and Methods section. Evaluation of the HO-1 level in nuclear fraction (**A**) and those of HO-1, MnSOD, MMP2, and VEGF in total fraction (**B**) by Western blotting analysis. Quantitative estimations of the protein levels were determined by densitometry measurements of western blotting from three independent experiments after normalization with β-actin. (*) *p* < 0.05, (**) *p* < 0.01 compared to the untreated sample; (^#^) *p* < 0.05; (^##^) *p* < 0.01 between the two groups.

**Figure 9 cancers-11-00505-f009:**
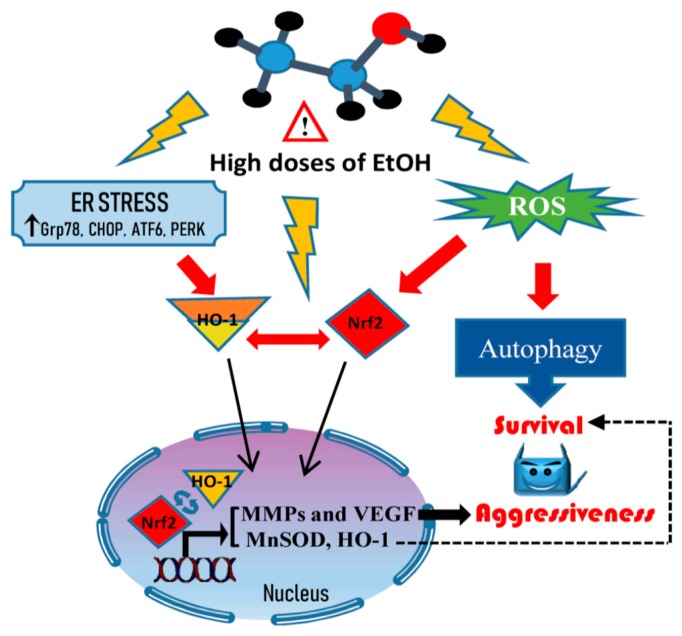
Scheme of the mechanism activated by high doses of ethanol. High doses of ethanol induce ROS generation and ER stress responsible for the induction of a prosurvival autophagic process and the aggressive tumor phenotype. Such events are sustained by the nuclear translocation of Nrf2 and HO-1, which activate antioxidant response systems and promote the upregulation of MMP2 and VEGF.

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
