# Peer review of "Ethanol-Mediated Stress Promotes Autophagic Survival and Aggressiveness of Colon Cancer Cells via Activation of Nrf2/HO-1 Pathway"

_cancers, 2019, doi:10.3390/cancers11040505_

Reviewer 1 Report

It is a very nice paper. It is well written and I suggest it for pubblication.

Author Response

Reviewer 1 comments It is a very nice paper. It is well written and I suggest it for publication.

We thank the Reviewer for appreciating the paper

Reviewer 2 Report

Comments:

    In this manuscript, the authors described “Ethanol-mediated stress promotes autophagic survival and aggressiveness of colon cancer cells via activation of Nrf2/HO-1 pathway”. This study reveals that EtOH mediates both the activation of Nrf2 and HO-1 to sustain colon cancer cell survival, thus leading to the acquisition of a more aggressive phenotype. The work seems not well-done in the area. However, several points need to be clarified.

Comments for the Attention of the Author(s)

There are some points should be checked are followings:

1.     The figure legends could be more concise and informative.  For example, doze, time point and sample size should be included.

2.     Western blots must demonstrate a representative molecular weight marker.

3.     Ethanol effect on colon cancer cell viability had some questions at 60 and 120 mM and different times. Please clarified it.

4.     Figure 2B, COX-2 protein densitometric analysis had some questions. Please clarified it.

5.     In the figure 2, Cellular Response to ER Stress. ER stress, initiated by accumulation of misfolded proteins, triggers activation of the ER transmembrane proteins PERK, ATF6 and IRE1. How about PERK, ATF6 and IRE1 protein expressions in the ethanol-mediated stress of colon cancer cells.

6.     In the results and discussion: The authors should strengthen the research’s results in the mechanism fields. After additional experiments and strengthening their description, this paper can be published.

Author Response

Comments 

1. The figure legends could be more concise and informative. For example, doze, time point and sample size should be included.

We thank the Reviewer for this comment. We modified the legends in accordance to the Reviewer’s request.

2. Western blots must demonstrate a representative molecular weight marker.

We thank the Reviewer for this comment. In each western blotting analysis we usually insert a pre-stained standard markers with a defined molecular weight. Since these markers are not chemiluminescent and are not visible during detection, we use to make small signals in correspondence to the markers before the immunodetection. Therefore, considering your reasonable request, we included the molecular weights in the Figures of western blotting analyses and these changes have been reported in the revised version of the manuscript.

3. Ethanol effect on colon cancer cell viability had some questions at 60 and 120 mM and different times. Please clarified it.

We thank very much the Reviewer’s perceptive comment which helped us to improve the reported information. In light of these observations, to clarify whether the reduction of cell viability observed with 60 and 120 mM at 24 and 48 h of ethanol treatment was due to cell death induction, we stained the cells with propidium iodide, a cell membrane impermeant dye that is generally excluded from viable cells. The results showed that ethanol treatment did not induce cell death also in the condition in which we observed a reduced metabolic utilization of MTT in the cell viability assay. Therefore, we hypothesize that such an effect observed in the first phase of treatment (24-48 h) with low doses of ethanol might be a consequence of a slowed metabolic cell activity or a cell cycle arrest. These observations were reported in the new version of the manuscript (see page 3, lines 107-112).

4. Figure 2B, COX-2 protein densitometric analysis had some questions. Please clarified it.

We thank the Reviewer for this comment. The densitometric analysis of COX-2 protein reported in Fig. 2B was calculated as the mean of three independent experiments. This could explain the lack of correspondence between COX-2 western blotting and its densitometric analysis.

5. In the figure 2, Cellular Response to ER Stress. ER stress, initiated by accumulation of misfolded proteins, triggers activation of the ER transmembrane proteins PERK, ATF6 and IRE1. How about PERK, ATF6 and IRE1 protein expressions in the ethanol-mediated stress of colon cancer cells.

We thank the Reviewer who suggested to include such a novel and interesting topic. In accordance with this observation, we evaluated the effect of ethanol on ATF6 and PERK. Our results demonstrated the activation of both these factors under ethanol-induced ER stress of colon cancer cells. We added these results in the new version of the manuscript by discussing their role in ethanol-induced cell survival (see page 5, lines 173-180; page 12, lines 353-354; page 12, lines 375-378, Figure 2C).

6. In the results and discussion: The authors should strengthen the research’s results in the mechanism fields.

We thank very much the Reviewer for this comment. In light of these observations, we made an effort to improve the manuscript in the mechanism fields providing a more detailed description of the cellular scenario under high exposure to ethanol. In particular, our analyses sustained an involvement of Atg7 in the autophagic process and an evident participation of ATF6 and PERK factors as main players in the ER stress activated by ethanol treatment. Such an effect reported in the results was also discussed along the text in the discussion section.

Reviewer 3 Report

Specific comments to the authors:

The authors Cesare Cernigliaro et al. investigated the role of Nrf2/HO-1 axis in relation to colon cancer survival and progression under ethanol (EtOH) stimulation in detail. Based on their applied broad range of molecular techniques the authors could demonstrate that EtOH could induce a more aggressive phenotype of colon cancer cells via the activation of Nrf2 and HO-1 to sustain colon cancer cell survival.

Overall, the manuscript is well done, easy to follow and to understand. The methods are overall well described. Although the results are clear presented, some minor concerns limit the impact of the manuscript (see specific comments). Finally, the discussion should discuss the limitation of the study mentioned in the specific comments in detail.

In conclusion, the presented data are very interesting. After incorporating the mentioned specific comments (see below) the manuscript has the potency to be accepted.

Specific comments

Abstract: Please add a definitive conclusion of the findings at the end of the abstract related to epidemiological and clinical aspects.

Results: The intensity and contrast of figure 2a and 3a could be enhanced.

Discussion: The authors should discuss the limitation that the findings based on protein-expression-data and an in-vitro-modell and were not transferred to animal-model and in-situ-analysis. Please add following relevant literature:

# Hepatology. 2014 Apr;59(4):1381-92.

# Food Funct. 2018 Aug 15;9(8):4184-4193.

# Neoplasma. 2010;57(1):86-92.

Presentation: Overall, the presentation is very good.

Author Response

1. Abstract: Please add a definitive conclusion of the findings at the end of the abstract related to epidemiological and clinical aspects.

We thank the Reviewer for this comment. Considering the overall study reported in our manuscript, we included a sentence on the epidemiological studies in the abstract. However, because of the restriction of words for this section, we chose report a more detailed description of the epidemiological studies which support the relationship between doses of alcohol consumption and CRC risk in the Introduction (see page 2, lines 52-58).

2. Results: The intensity and contrast of figure 2a and 3a could be enhanced.

We thank the Reviewer for this comment. We modified figures 2A and 2B by enhancing the intensity and contrast.

3. Discussion: The authors should discuss the limitation that the findings based on protein-expression-data and an in-vitro-model and were not transferred to animal-model and in-situ-analysis.

We thank the Reviewer for this comment. According to the reviewer’s request, we discussed the limitation of the findings based on an in-vitro model by adding a new sentence in the final part of the discussion ( see page 13 lines 416-421).

Please add following relevant literature:# Hepatology. 2014 Apr;59(4):1381-92.;  Food Funct. 2018 Aug 15;9(8):4184-4193; Neoplasma. 2010;57(1):86-92.

We added the suggested papers in the new version of the manuscript (references 50; 18; 69)

Reviewer 4 Report

This paper attempts to consider the signalling pathways that promote cell survival in response to extremely high concentrations of alcohol when applied to different colorectal cancer cell lines.  There are many fundamental flaws in the manuscript writing as well as the experimental design.  The concentrations of alcohol applied and the time points considered are not physiological nor pathologically relevant.  300 mM ethanol will often kill humans, so to apply levels of 100mM-300mM directly to cells and for exposure periods of 1-3 days has little experimental relevance.  Nevertheless, high concentrations of alcohol may facilitate assessment of the mechanism of survival, but the experimental data presented is of poor quality.

The authors make a series of bold statements:

Abstract: strongly associated with colorectal cancer – what are the relative risks provided by epidemiological studies?  High risk is more for high alcohol consumers.

Intro: ROS generated by CYP2E contributes to colon carcinogenesis – how has this been proven?

Figure 1: comment on why the ‘cell viability’ does decline initially in HT29 and Caco-2 cells.  Also MTT assays measure metabolic activity not cell viability, but can be a surrogate for cell viability if additional measures are made such as LDH or ATP levels.

Comment also that the applied concentrations are ridiculously high, and furthermore, even with individuals that experience 300 mM plasma concentrations, they will be undergoing metabolism of alcohol.  These cells will not.  Secondly, when incubating cells with such high concentrations of alcohol, how can the authors account for evaporation of the agent over time?

The blotting shown is of poor quality – with only 3 experiments performed.  The authors provide no basis for using β-actin – or demonstration that levels of β-actin do not change under the conditions employed.  What about whole lane protein staining for normalisation.

Why not use primary cells to demonstrate cell survival and potential for transformation to a cancerous phenotype?  These are already cancer cells.  The authors state leading to a more aggressive phenotype in the abstract but this has not been shown.

Figure 3 blots – very poor.

Figure 4 – differences of say 4-fold are claimed for MnSOD – although this is not visually apparent from the western blots.  If this is so, substantiate the results with activity assay measurements.

Likewise, Figure 6 blots – 7-fold changes – really?  Demonstrate this with activity measurements.

The autophagic/mitophagic effects of ethanol are often demonstrated visually using EM.

Overall, the manuscript lacks theoretical and experimental validation and direction.  I would suggest the authors consider the comments outlined above before proceeding further with the work.

Author Response

Abstract: strongly associated with colorectal cancer – what are the relative risks provided by epidemiological studies? High risk is more for high alcohol consumers.

We thank the Reviewer for this comment. Considering the overall study reported in our manuscript, we included a sentence on the epidemiological studies in the abstract. However, because of the restriction of words for this section, we chose to report a more detailed description of the epidemiological studies which support the relationship between doses of alcohol consumption and CRC risk in the Introduction (see page 2, lines 52-58).

Intro: ROS generated by CYP2E contributes to colon carcinogenesis – how has this been proven?

We thank the Reviewer for this comment. Accordingly, we clarified in the Introduction the role of CYP2E1 overexpression in colon carcinogenesis and its relationship with ROS generation observed in conditions of chronic alcohol consumption (see page 2, lines 66-70).

Figure 1: comment on why the ‘cell viability’ does decline initially in HT29 and Caco-2 cells. Also MTT assays measure metabolic activity not cell viability, but can be a surrogate for cell viability if additional measures are made such as LDH or ATP levels.

We thank very much the Reviewer’s perceptive comment which helped us to improve the reported information. In light of these observations, to clarify whether the reduction of cell viability observed with 60 and 120 mM at 24 and 48 h of ethanol treatment was due to cell death induction, we stained the cells with propidium iodide, a cell membrane impermeant dye that is generally excluded from viable cells. The results showed that ethanol treatment did not induce cell death also in the condition in which we observed a reduced metabolic utilization of MTT in the cell viability assay. Therefore, we hypothesize that such an effect observed in the first phase of treatment (24-48 h) with low doses of Ethanol might be  a consequence of a slowed metabolic cell activity or a cell cycle arrest. These observations were reported in the new version of the manuscript (see page 3, lines 107-112).

Comment also that the applied concentrations are ridiculously high, and furthermore, even with individuals that experience 300 mM plasma concentrations, they will be undergoing metabolism of alcohol. These cells will not  Secondly, when incubating cells with such high concentrations of alcohol, how can the authors account for evaporation of the agent over time?

We thank the Reviewer for this comment. According to the epidemiological data present in the literature, 50 g/day is considered a high concentration of ethanol which is associated with high risk of developing colon cancer. However, it is difficult to compare this dose with the molar concentration present in the plasma after ethanol ingestion although, as reported by Singletary (Singletary et al. Cancer 2001, 165, 131–137), 100 mM represents a dose of ethanol in the plasma corresponding to high ethanol consumption. It should also be considered that colonocytes come in contact directly with the ingested ethanol, before it is diluted in the plasma. Finally, we underline that our study is performed in vitro and therefore we used ethanol concentrations which are reported in literature to assess the effect of ethanol on cultured cells. The high doses used (100-300 mM) are toxic in different cell lines of other cancer types (Haorah, J.; Free Radical Biology and Medicine 2008, 45, 1542–1550; Kawaratani, H.; Mediators of Inflammation 2013, 2013, 1–10)  but they do not exert toxic effects in colon cancer cells as also shown in both in vitro and in vivo studies (Xu M. , Mol Carcinog 2016, 55:1002-1011; Banan, The J Pharmacol Exp Therapeutics, 291:1075–1085, 1999). Moreover, considering that ethanol is a volatile alcohol, ethanol-containing medium was substituted daily. This information has been added in the new version of the manuscript (see page 14, lines 445-446).

The blotting shown is of poor quality – with only 3 experiments performed.  The authors provide no basis for using β-actin – or demonstration that levels of β-actin do not change under the conditions employed.  What about whole lane protein staining for normalization.

We thank the Reviewer for this comment. The change in protein level by western blotting is normally considered significant by analysing three different experiments. Moreover, β-actin was used as a loading control to normalize the data, after checking that β-actin content was not modified by ethanol treatment vs untreated sample. The positive results permits us to use b-actin as loading control, such as largely employed. This is in accordance with the observation of Lamichhane et al. Sci. Rep. 2017, 7. This aspect was also discussed in methods section. (see page 15, lines 477-480).

Why not use primary cells to demonstrate cell survival and potential for transformation to a cancerous phenotype? These are already cancer cells. The authors state leading to a more aggressive phenotype in the abstract but this has not been shown.

We thank the Reviewer for this comment. We performed the study on colon cancer cells because we aim to clarify the biochemical mechanisms underlying the effect of ethanol on the progression of CRC. To this regard, the ethanol-induced activation of metalloproteases, demonstrated by gel zymography, can represent a mechanism to favour tumour progression.

Figure 3 blots – very poor. We thank the Reviewer who suggested to include additional data regarding the activation of autophagy. According with the reviewer’s request, we evaluated the effect of Ethanol treatment on the level of ATG7, another autophagy marker which was involved in the described mechanism. These new results were added in the new version of the manuscript (see page 6, lines 200-202, Figure 3).

Figure 4 – differences of say 4-fold are claimed for MnSOD – although this is not visually apparent from the western blots. If this is so, substantiate the results with activity assay measurements.

We thank the Reviewer for this comment. We double-checked the densitometric analyses of MnSOD. The data confirm a 4-fold increase in the level of this antioxidant enzyme after ethanol treatment, which justifies a rapid decrease in ROS levels. As far MnSOD activity we are going to explore these aspects in future studies.

Likewise, Figure 6 blots – 7-fold changes – really? Demonstrate this with activity measurements.

We thank the Reviewer for this comment. The 7-fold changes in the level of MMP-2 and MMP-9 observed in ethanol-treated HCT116 colon cancer cells are referred to the cleaved and active forms of these enzymes. Moreover, activation of both MMP2 and MMP9 has been demonstrated by gel zymography, which evaluates the ability of these gelatinases to degrade the gelatin-impregnated gel as consequence of their activation.

The autophagic/mitophagic effects of ethanol are often demonstrated visually using EM.

We thank the Reviewer for this comment. We know that autophagy can be demonstrated visually using EM and the criticism of the reviewer of this subject is certainly appropriated . We have in mind to develop this aspect in the future also to clarify possible mitophagic processes activated by authophagy and the mechanisms through which this event could occur in our conditions. Differently, this study aimed at demonstrating ethanol-mediated activation of autophagy by fluorescence microscopy employing monodansylcadaverine (MDC), a fluorescent marker which is employed to detect the formation of autophagic vacuoles. To support the involvement of autophagic flux our analyses also provided evidence of the increased level of  lipidated form LC3 (LC3-II) in ethanol-treated samples as well as an increase in Atg7 which was included in the revised version of the manuscript.

Round  2

Reviewer 2 Report

Figure 2. Ethanol induces oxidative and ER stress in colon cancer cells. Data are missing.

Figure 9. Scheme of the mechanism activated by high doses of ethanol. The ER stress, MMPs  and Nrf2/HO-1 proteins shall be includings.

Author Response

Reviewer 2 comments.  

Figure 2. Ethanol induces oxidative and ER stress in colon cancer cells. Data are missing.

We thank the Reviewer for this comment. We apologize, but Figure 2 regarding  the effect of ethanol in oxidative and ER stress in colon cancer cells did not appear in the PDF version of the revised manuscripit. Now we uploaded a new version of the manuscript containing the Figure 2.

Figure 9. Scheme of the mechanism activated by high doses of ethanol. The ER stress, MMPs and Nrf2/HO-1 proteins shall be includings.

We really appreciated this suggestion and modified the Figure 9 by evidencing the role of ER stress and its markers involved, as well as MMPs and Nrf2/HO-1 in the mechanism activated by high doses of ethanol.

Reviewer 4 Report

The authors have performed a good revision of the manuscript and added new supporting work as well as clarified areas of concern.  The manuscript is suitable for publication.

Author Response

Reviewer 4 comments.

The authors have performed a good revision of the manuscript and added new supporting work as well as clarified areas of concern. The manuscript is suitable for publication.

We thank the reviewer for appreciating our efforts to ameliorate the revised version of the manuscript.